# A broadly neutralizing monoclonal antibody overcomes the mutational landscape of emerging SARS-CoV-2 variants of concern

Hilal Ahmad Parray[1☉], Naveen Narayanan[2☉], Sonal Garg[1☉], Zaigham Abbas Rizvi[1☉], Tripti Shrivastava[1], Sachin Kushwaha[3], Janmejay Singh[1], Praveenkumar Murugavelu[1], Anbalagan Anantharaj[1], Farha Mehdi[1], Nisha Raj[1], Shivam Singh[1], Jyotsna Dandotiya[1], Asha Lukose[1], Deepti Jamwal[1], Sandeep Kumar[1], Adarsh K. Chiranjivi[1], Samridhi Dhyani[1], Nitesh Mishra[4], Sanjeev Kumar[5], Kamini Jakhar[1], Sudipta Sonar[1], Anil Kumar Panchal[1], Manas Ranjan Tripathy[1], Shirlie Roy Chowdhury[1], Shubbir Ahmed[1], Sweety Samal[1], Shailendra Mani[1], Sankar Bhattacharyya[1], Supratik Das[1], Subrata Sinha[4], Kalpana Luthra[4], Gaurav Batra[1], Devinder Sehgal[3], Guruprasad R. Medigeshi[1], Chandresh Sharma[1]*, Amit Awasthi[1]*, Pramod Kumar Garg[1‡]*, Deepak T. Nair[2‡]*, Rajesh Kumar[1¤‡]*

1 Translational Health Science and Technology Institute, NCR Biotech Science Cluster, Faridabad, Haryana, India, 2 Regional Centre for Biotechnology, NCR Biotech Science Cluster, Faridabad, Haryana, India, 3 National Institute of Immunology, Aruna Asaf Ali Marg, New Delhi, India, 4 Department of Biochemistry, All India Institute of Medical Sciences, New Delhi, India, 5 ICGEB-Emory Vaccine Center, International Centre for Genetic Engineering and Biotechnology, New Delhi, India

☉ These authors contributed equally to this work.
¤ Current address: Institute of Advanced Virology (IAV), Trivandrum, Kerala, India
‡ Lead Corresponding authors
* chandresh@thsti.res.in (CS); aawasthi@thsti.res.in (AA); pgarg@thsti.res.in (PKG); deepak@rcb.res.in (DTN); rajesh@thsti.res.in (RK)

**Data Availability Statement:** The coordinates of the structure of P4A2 Fab in complex with Spike-RBD have been deposited in the Protein Data Bank

## Abstract

The emergence of new variants of SARS-CoV-2 necessitates unremitting efforts to discover novel therapeutic monoclonal antibodies (mAbs). Here, we report an extremely potent mAb named P4A2 that can neutralize all the circulating variants of concern (VOCs) with high efficiency, including the highly transmissible Omicron. The crystal structure of the P4A2 Fab: RBD complex revealed that the residues of the RBD that interact with P4A2 are a part of the ACE2-receptor-binding motif and are not mutated in any of the VOCs. The pan coronavirus pseudotyped neutralization assay confirmed that the P4A2 mAb is specific for SARS-CoV-2 and its VOCs. Passive administration of P4A2 to K18-hACE2 transgenic mice conferred protection, both prophylactically and therapeutically, against challenge with VOCs. Overall, our data shows that, the P4A2 mAb has immense therapeutic potential to neutralize the current circulating VOCs. Due to the overlap between the P4A2 epitope and ACE2 binding site on spike-RBD, P4A2 may also be highly effective against a number of future variants.

## Author summary

Existing SARS-CoV-2 vaccines and antibody therapies may be rendered ineffective by newly developing SARS-CoV-2 mutations which are exceedingly alarming. This means

with accession code 7WVL. https://www.rcsb.org/structure/unreleased/7WVL;Author.

**Funding:** The work for isolation and characterization of mAbs was supported by the Translational Health Science and Technology Institute Intramural grant (Grant No. THSTI-T001 to RK) and GIISER South Asia grant from Bill and Melinda Gates Foundation, Seattle, USA (Grant No. Investment INV-030592 to PKG). Intramural funds from the Regional Centre for Biotechnology (RCB) were utilized for structural work in this study (Grant No. RCB- C01001 to DTN). THSTI and RCB are funded by the Department of Biotechnology, Ministry of Science and Technology, Government of India. Data collection at the ESRF was supported by the ESRF Access Program of the Department of Biotechnology (Grant No. BT/PR36150/INF/22/214/2020). Animal studies work was supported by financial support from BIRAC (Grant No. BT/CS0054/05/21 to AA). The funders had no role in study design, data collection and analysis, decision to publish, or preparation of the manuscript.

**Competing interests:** I have read the journal's policy and the authors of this manuscript have the following competing interests. THSTI has filled a provisional patent application in India (Indian Patent Application No. 202211005568; Filed on: February 2, 2022).

that existing vaccines and antibody-based countermeasures must be thoroughly assessed against the mutational variants that already exist in order to determine their protective efficiency. In this study, we isolated a highly potent and broadly neutralizing murine monoclonal antibody having broad neutralizing efficacy against SARS-CoV-2 and its variants of concern (VOCs). Structural characterization of P4A2 Fab in complex with the receptor-binding domain defined a substantially overlapped surface required for human angiotensin-converting enzyme-2 (hACE2) interaction. The SARS-CoV-2 VOCs challenge paradigm in K18-hACE2-transgenic mice against SARS-CoV-2 VOCs, revealed the broad and potent effects of the leading clone, P4A2, both prophylactically and therapeutically *in vivo*. Additionally, humanized P4A2 mAb may be used alone or in conjunction with other non-competing antibodies as an effective cocktail method for SARS-CoV-2 prophylactics or therapeutics, or additives to current cocktails to safeguard against emerging variants. Our findings provide insights into the design of biopharmaceuticals against current and future emerging variants.

## Introduction

Existing SARS-CoV-2 vaccines and antibody therapies are rendered less effective or ineffective by newly emerging SARS-CoV-2 mutations, escalating the threat towards the human race. A recent study demonstrated that the majority of monoclonal antibodies (mAbs) directed against receptor-binding domain (RBD) of the SARS-CoV-2 spike protein exhibited a considerable reduction in the *in vitro* neutralizing activity against the Omicron variant [1,2]. It underscores the urgent need for the development of novel broadly neutralizing antibodies against the emerging variants of concern (VOCs).

## Results and discussion

In the present study, we generated a panel of anti-SARS-CoV-2 mAbs using mouse hybridoma technology, by immunizing BALB/c mice with purified RBD protein that corresponds to Alpha variant of SARS-CoV-2 (N501Y). The neutralization specificity of the hybridoma supernatants was screened, one of the mAb, P4A2, demonstrated exceptionally efficient and broad neutralization of both ancestral SARS-CoV-2 WA1/2020 and other VOCs, Alpha, Beta, Kappa and Delta (range = 10–39 ng mL$^{-1}$ corresponding to 0.07 to 0.26 nM) (Fig 1A). The P4A2 binds with RBDs of different VOCs with high affinity, and binding assays based on competition suggested that P4A2 ACE2 binding sites overlap (Figs 1B and S1).

 The crystal structure of P4A2 Fab in complex with the RBD-N501Y (residues 332–544) of protein was determined to a maximum resolution of 3.0 Å (Fig 1C). The crystal structure shows the electron density for the entire heavy and light chain of the P4A2 Fab. For the RBD, the first five residues (332–336) was disordered, and the density for residues 514–544 at the C-terminal is missing. The structure showed that the residues 475–489 and 455–456 of the RBD are present close to the P4A2 Fab paratope (Fig 1D). The residues of the RBD that interact with the paratope of the P4A2 Fab are 455Leu, 456Phe, 483Val, 484Glu, 485Gly, 486Phe, 487Asn and 489Tyr (Fig 1D). The P4A2 paratope is made up of residues T30, R31, Y32, S33, Y35 and M37 from CDR1 along with N52 from CDR2 and S99 from CDR3 of the heavy chain. The residues involved from CDR1 of the light chain are Y31, T34, L36, Q38 and F40, and that from CDR2 are Y53, A54 and N57. Q93 and S95 from CDR3 of the light chain also contribute to the paratope (Fig 1D). The area of the surface buried due to interaction between RBD and P4A2 paratope is 1667 Å$^2$. The key interactions that stabilize the distributed epitopes from the

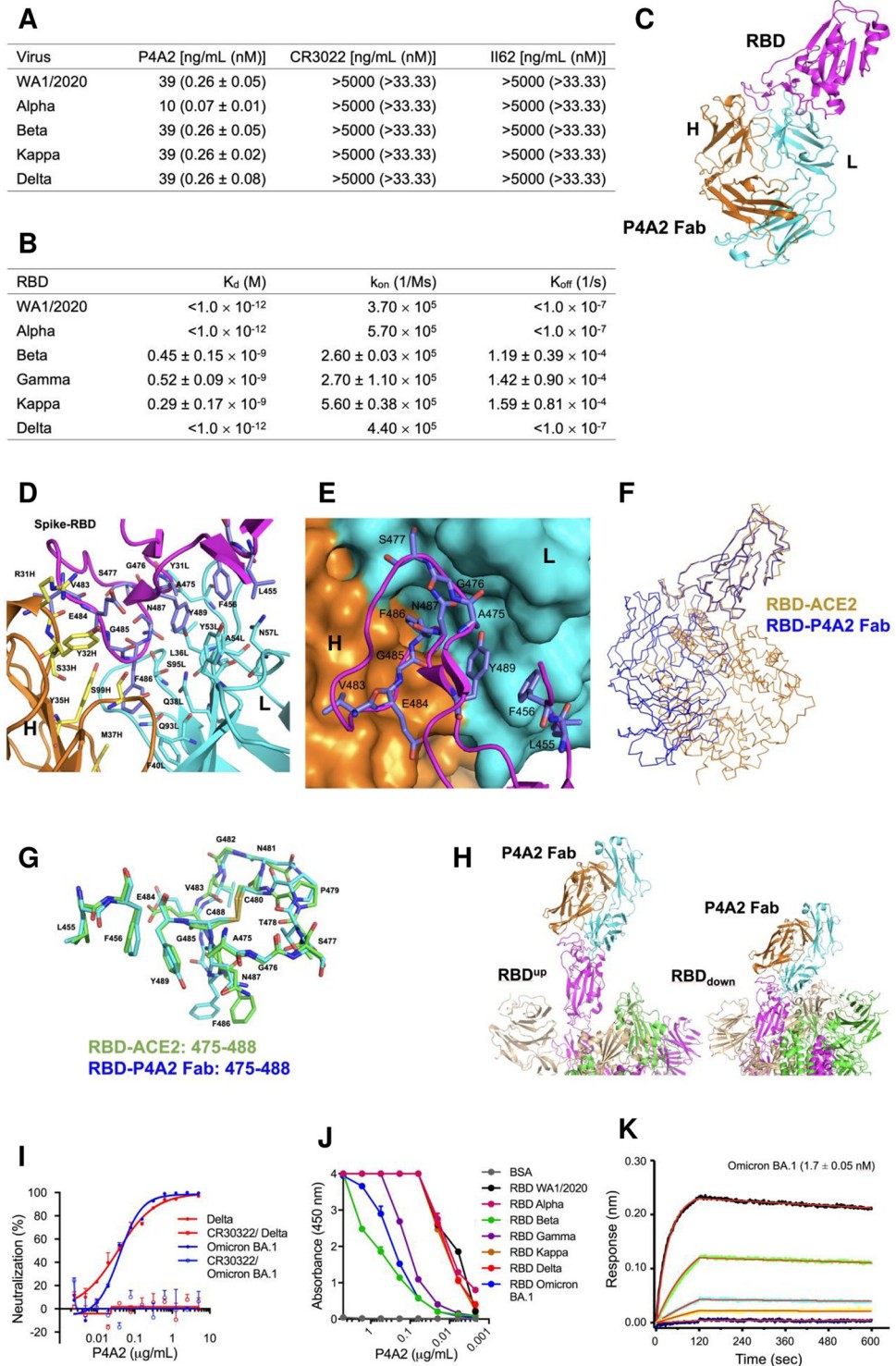

**A**

| Virus | P4A2 [ng/mL (nM)] | CR3022 [ng/mL (nM)] | Il62 [ng/mL (nM)] |
|---|---|---|---|
| WA1/2020 | 39 (0.26 ± 0.05) | >5000 (>33.33) | >5000 (>33.33) |
| Alpha | 10 (0.07 ± 0.01) | >5000 (>33.33) | >5000 (>33.33) |
| Beta | 39 (0.26 ± 0.05) | >5000 (>33.33) | >5000 (>33.33) |
| Kappa | 39 (0.26 ± 0.02) | >5000 (>33.33) | >5000 (>33.33) |
| Delta | 39 (0.26 ± 0.08) | >5000 (>33.33) | >5000 (>33.33) |

**B**

| RBD | $K_d$ (M) | $k_{on}$ (1/Ms) | $K_{off}$ (1/s) |
|---|---|---|---|
| WA1/2020 | $<1.0 \times 10^{-12}$ | $3.70 \times 10^5$ | $<1.0 \times 10^{-7}$ |
| Alpha | $<1.0 \times 10^{-12}$ | $5.70 \times 10^5$ | $<1.0 \times 10^{-7}$ |
| Beta | $0.45 \pm 0.15 \times 10^{-9}$ | $2.60 \pm 0.03 \times 10^5$ | $1.19 \pm 0.39 \times 10^{-4}$ |
| Gamma | $0.52 \pm 0.09 \times 10^{-9}$ | $2.70 \pm 1.10 \times 10^5$ | $1.42 \pm 0.90 \times 10^{-4}$ |
| Kappa | $0.29 \pm 0.17 \times 10^{-9}$ | $5.60 \pm 0.38 \times 10^5$ | $1.59 \pm 0.81 \times 10^{-4}$ |
| Delta | $<1.0 \times 10^{-12}$ | $4.40 \times 10^5$ | $<1.0 \times 10^{-7}$ |

**Fig 1. Biochemical and structural analysis reveals that P4A2 mAb can neutralize all circulating variants of concern.** (A) Neutralization of authentic SARS-CoV-2 VOCs by P4A2 was determined using virus-induced cytopathic effect (CPE) based assay, neutralization values are shown in ng/mL and nmol. (B) Kinetics of P4A2 binding to RBDs from VOCs was assessed by Bio-Layer Interferometry (BLI) Octet. (C) Structure of P4A2 Fab in complex with RBD of Alpha variant. The heavy (H) and light (L) chain of P4A2 Fab are coloured in orange and cyan, respectively and the RBD is coloured magenta. (D) The interacting residues from P4A2 paratope and the RBD epitope are displayed in stick representation and coloured according to element. The carbon atoms of H chain, L chain and RBD are coloured

in yellow, cyan and magenta, respectively. (E) Surface representation of the P4A2 paratope with the RBD epitope is shown. 486Phe from RBD is present in a hydrophobic cavity formed on the paratope. (F) Superimposition of the RBD-P4A2 Fab (blue) and the RBD-ACE2 structures (brown) shows that P4A2 binding to RBD will prevent interaction of the viral protein with the ACE2 receptor. (G) Superimposition of the structure of residues 475–488 and 455–456 of RBD when bound to P4A2 Fab and ACE2 receptor. The carbon atoms of this stretch when bound to Fab and ACE2 are coloured cyan and green, respectively. (H) Computational model of P4A2 Fab bound to spike trimer shows that the P4A2 can bind to the RBD of the trimer in both the ACE2 receptor accessible "up" and receptor-inaccessible "down" positions. (I) Neutralization of authentic SARS-CoV-2 BA.1 and Delta by P4A2 was determined using focus reduction neutralization assay, CR3022 mAb was used as experimental negative control, the CR3022 is a potent neutralizing mAb for SARS-CoV-1 and shows non-neutralizing behaviour against SARS-CoV-2. Focus reduction neutralization assays were done in duplicates and repeated at least three times. (J) Binding avidity ($EC_{50}$) of P4A2 to RBD proteins of different VOCs i. e. WA1/2020, Alpha, Beta, Gamma, Kappa, Delta and BA.1 were determined by ELISA, the $EC_{50}$ values are 0.0093, 0.0099, 0.6002, 0.1158, 0.0131, 0.0121 and 0.2616, respectively. (K) Kinetics of P4A2 binding to BA.1 RBD protein was assessed by BLI-Octet. For reproducibility of data, all the P4A2 mAb characterization experiments were repeated at least three independent times in triplicates.

receptor-binding motif (RBM) in the P4A2 paratope include (i) the presence of the aromatic ring of 486Phe deep inside a hydrophobic cavity (Fig 1E) lined by F40L, Y35H and M37H, (ii) hydrogen bond formed between backbone carbonyl of R31H and backbone nitrogen of 484Glu, (iii) hydrogen bond formed between backbone carbonyl of 484Glu and backbone nitrogen of S33H, (iv) hydrogen bond formed between backbone nitrogen of 486Phe and side chain of Y35H, (v) hydrogen bond formed between backbone carbonyl of 485Gly and side chain of S99H, (vi) hydrogen bond formed between side chain of 487Asn and backbone carbonyl of S95L and (vii) hydrophobic interactions formed between 456Phe, 489Tyr, L36L and Y53L.

Some of the residues of RBD that interact with the P4A2 paratope are a part of the RBM that interact with the human ACE2. Based on the crystal structure of the SARS-CoV-2 RBD in complex with human ACE2 (PDB code: 6M0J), the viral protein forms interactions with the human receptor through the following residues: 417Lys, 446Gly, 449Tyr, 453Tyr, 455Leu, 456Phe, 475Ala, 486Phe, 487Asn, 489Tyr, 493Gln, 496Gly, 498Gln, 500Thr, 501Asn, 502Gly and 505Tyr [3]. Among these residues 455Leu, 456Phe, 486Phe, 487Asn and 489Tyr form key hydrophobic and polar interactions with the paratope of P4A2 Fab and therefore, binding of the Fab to the spike-RBD will render these residues inaccessible to the ACE2 receptor. A superimposition of the RBD from the complex with P4A2 Fab onto that from the complex with ACE2 shows that the viral protein bound to P4A2 will be unable to engage with the human receptor due to steric clashes (Fig 1F). The backbone conformation of the stretch spanning residues 475–488 and 455–456 from spike-RBD is similar when bound to P4A2 and to ACE2 (6M0J) with an RMSD of 0.8 Å (Fig 1G). Except for 486Phe which exhibits a different side chain conformation, there is a substantial overlap in the side chain orientation for the other residues. The structure of the P4A2:Fab complex was utilized to generate two computational models of the Fab in complex with spike trimer which showed that P4A2 should be able to bind to the RBD when it is both in the "up" (accessible to ACE2 receptor) or "down" (inaccessible to ACE2 receptor) positions (Fig 1H). Overall, the composition and conformation of the epitope in RBD and the mode of binding of the Fab ensures that P4A2 interaction will prevent recognition of ACE2 by the spike protein and thus prevent entry of the virus into the host cell.

The computational models of P4A2 Fab in complex with the spike-RBD from Beta, Gamma, Delta, Kappa and Omicron (BA.1) VOCs were generated. These models show that, for these VOCs, there are no mutations in the residues that interact with the P4A2 through their side-chain. E484 is mutated to Lys, Gln or Ala in some of the VOCs, but this residue forms interactions with the P4A2 Fab paratope through the backbone atoms and not through the side chain.

Overall, the structural analysis provides an explanation regarding the ability of the P4A2 mAb to neutralize the Alpha, Beta, Gamma, Kappa and Delta variants, and also suggests that the mutations in residues of the RBD observed in the BA.1 variant will not abrogate P4A2: spike protein interaction (S2 Fig). To validate this inference, we tested the neutralization potential of P4A2 with live BA.1 virus. The P4A2 neutralized BA.1 with an $IC_{50}$ of 45 ng mL$^{-1}$ (0.3 nM) (Fig 1I), and binds to purified BA.1 RBD with high specificity and nanomolar affinity (Fig 1J and 1K). The P4A2 shows strong binding specificity in ELISA with half-maximal effective concentration ($EC_{50}$) of 0.001–0.600 ng/mL. These values were higher for P4A2 binding with Beta, Gamma and BA.1 RBD proteins as compared to RBD from other VOCs tested (S1B Fig). However, Bio-Layer Interferometry (BLI) data suggests that P4A2 binds with nanomolar to sub-nanomolar affinity to RBD proteins of all VOCs tested (Figs 1J and S1A). Further, the neutralizing activity of P4A2 was unchanged against all the VOCs tested, suggesting that slightly higher $EC_{50}$ binding values for Beta, Gamma and BA.1 RBD proteins does not have any negative effect on the P4A2 neutralization mechanism.

It is possible that mutation of the E484 residue to Lys, Lys or Ala may be responsible for the observed reduction in affinity in the case of RBD from Beta, Gamma and BA.1, respectively. From the crystal structure of spike-RBD:P4A2 Fab complex, it is clear that this residue forms interactions with the Fab paratope through the polypeptide backbone atoms. The side chain is not involved in these interactions, but the change in electrostatic surface of the molecule due to these mutations may increase the $K_{off}$ leading to a decrease in the equilibrium affinity. However, the affinity is still in the nanomolar range and therefore the E484 mutations have minimal effect on the ability of P4A2 to neutralize the SARS-CoV-2 virus in cell culture and animal models. This is further corroborated by immunofluorescence data that the number of foci recognized by P4A2 is similar in all the VOC-infected cells and reveals its broad reactivity to cells infected with all VOCs studied (Figs 2A, S3 and S4).

The broad-spectrum antiviral intervention effect of P4A2 against different Alpha and Beta coronaviruses was tested using vesicular stomatitis virus (VSV) pseudotyped viruses; MERS-CoV S, OC43 S, SARS-CoV S, SARS-CoV-2 S and HKU1 S, confirming that the epitope recognised by P4A2 is present only in the SARS-CoV-2 family and no cross-neutralization was seen with other coronaviruses (Fig 2B).

We next determined whether P4A2 could confer protection *in vivo* in a K18 hACE-2 mouse challenge model of SARS-CoV-2 infection. Eight to ten-week-old animals were administered a single dose (5 mg/kg) of P4A2 intraperitoneally to assess its prophylactic efficacy (against WA1/2020 SARS-CoV-2 and Kappa variant) and therapeutic effect (against WA1/2020 SARS-CoV-2, Kappa, Beta and Delta variant). Animals were challenged intranasally with $10^5$ plaque forming units (PFU) of the virus. In the prophylactic group, antibody was infused one day prior to virus challenge and in therapeutic group antibody was administrated 12 h following infection with the virus. The changes in body weight of experimental animals were monitored daily for 6 days. Additional clinical data was analysed in order to calculate the overall disease severity index at day 6 or 7 post-challenge. On day 6, mice in the infection control group lost more than 10% of their body weight compared to those in the P4A2 treatment group (Fig 2C). Furthermore, as compared to the virus challenge group, mice given P4A2 had significantly reduced viral RNA levels in their lungs ($p < 0.001$; Figs 2D, S5 and S6).

To determine the minimal protective dose for therapeutic protection, we passively administered P4A2 at a lower dose (1 mg/kg). A 20 µg dose of the P4A2 was fully protective and sufficient to suppress viral replication in the lungs, confirming the high potency of P4A2 *in vivo* against the Beta and Delta variant (Figs 2E, 2F, S5 and S7).

Few mAbs are currently available that effectively neutralize all VOCs [4]. Sotrovimab (S309) is reported to neutralize the BA.1 variant, but at a significantly lesser potency than

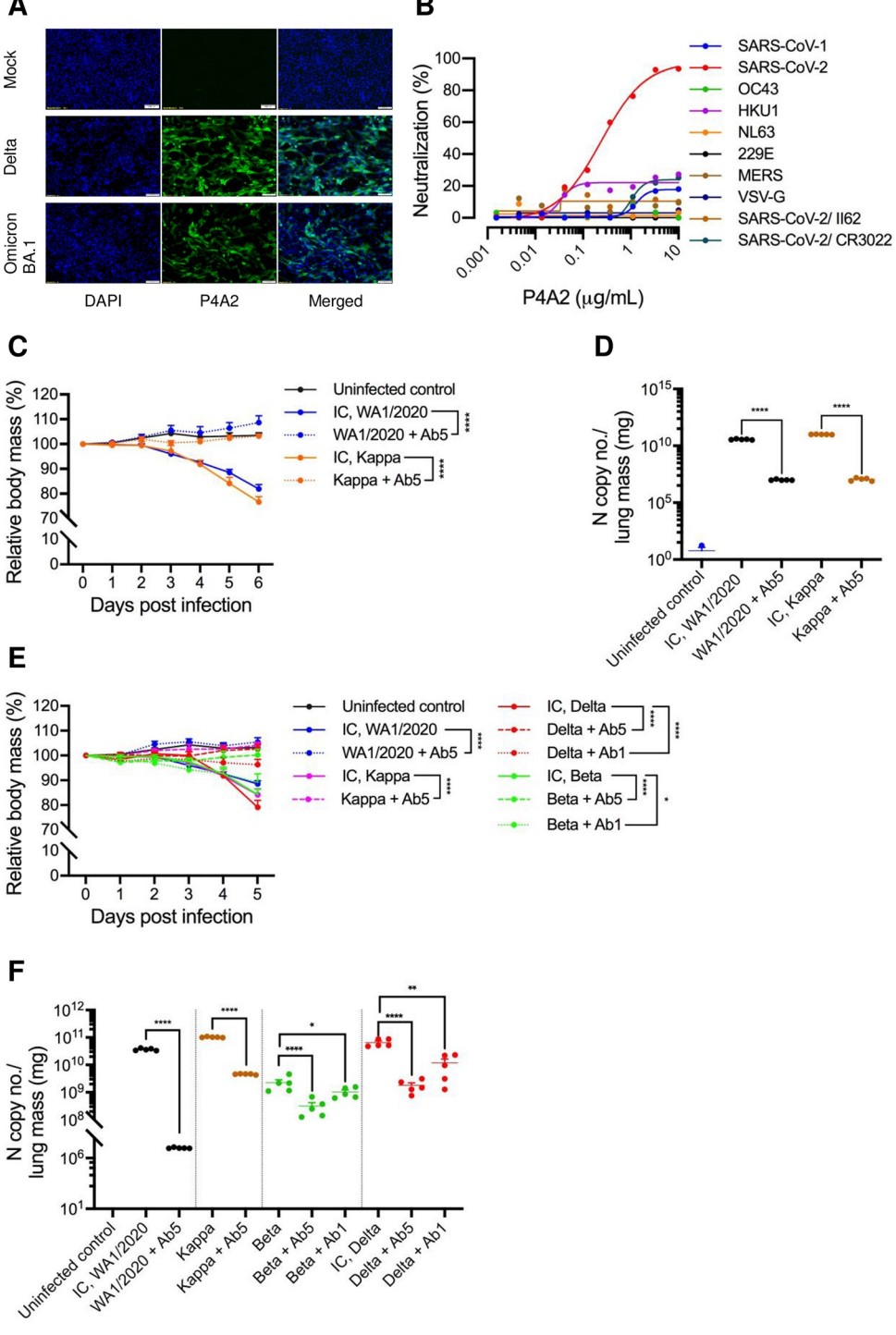

**Fig 2. P4A2 protects K18-hACE2 mice against SARS-CoV-2 VOCs.** (A) Calu-3 cells were infected with Delta and BA.1 variants of SARS-CoV-2 at a MOI of 0.5. At 24 h post infection, cells were fixed with chilled methanol, stained with P4A2 antibody and visualized by immunofluorescence microscopy. DAPI was used to stain nuclei. Images were captured at a magnification of 10×. Representative image of uninfected cells (*top row*), Calu-3 cells infected with Delta strain (*middle row*) and Calu-3 cells infected with BA.1 strain of SARS-CoV-2 (*bottom row*). Scale bar represents 100 μm. (B) Broad spectrum cross-neutralization potential of P4A2 against Alpha and Beta coronaviruses was assessed in pseudotyped viral neutralization assay, SARS-CoV-2 WA1/2020 strain was used in this assay. The P4A2 specifically neutralized SARS-CoV-2 with an $IC_{50}$ of 230 ng/mL. (C-F) P4A2 offers low-dose prophylactic and therapeutic protection in the K18-hACE2 mouse model. P4A2 antibody was infused intraperitoneally into mice as a single dose of

100 μg (5 mg/kg body weight) as prophylactic treatment, 24 h prior to intranasal inoculation with $10^5$ PFU of SARS-CoV-2 Wuhan and Kappa isolate. (C) Body mass of mice from each group was recorded for 6 days post infection. The Y-axis represents the body mass of the animal relative to the body mass of the same animal recorded on day 0 (normalised to 100) (dpi, days post-infection). (D) Lung RNA samples were used to evaluate viral load by qPCR for N gene against a known standard. The N gene copy number values obtained were plotted as bar graph as mean ± SEM. (E) In therapeutic treatment group, the mAb was administrated 12 h post-infection. Percent change in body mass of mice challenged with Wuhan, Kappa, Delta and Beta variant in the presence or absence of therapeutic treatment of P4A2 at two concentrations (5 and 1 mg of mAb per kg body mass) is plotted as a line graph till 6 dpi. 'IC' denotes infection control group, and Ab5 and Ab1 denotes 5 and 1 mg/kg dose of P4A2 mAb, respectively. The Y-axis in panel C and E represents percentage change in the body mass of the animal relative to the body mass recorded on day 0. (F) N gene copy number for viral load assessment from the lungs of the mice at 6 dpi. Data is represented as mean ± SEM values for each group. *, $p < 0.05$; **, $p < 0.01$; ***, $p < 0.001$ and ****, $p < 0.0001$ (One-way or Two-way ANOVA).

P4A2. Although sotrovimab and P4A2 binds with BA.1 RBD with similar affinity (Figs 3A–3C and S8). The observed difference in the neutralization potential of these two mAbs might be because they target different epitopes with different mechanisms of neutralization. The sotrovimab recognizes a glycan epitope, without competing with receptor attachment. However, P4A2 directly binds to the site that competes with virus attachment to host cells. This might be one probable reason that sotrovimab can neutralize the Omicron BA.1 variant, but with significantly lesser potency (S1 Table).

P4A2 exhibits strong interactions with residues of RBD that are critical for binding to the ACE2 receptor. Hence, the mutations in the RBD that reduce recognition by P4A2 may also plausibly adversely impact binding to the ACE2 receptor. Therefore, it is possible that this mAb may be able to neutralize new variants that may arise in the future. Maher et al. (2022) [5] predict that the following mutations will perturb the ability of therapeutic mAbs to recognize spike-RBD: A344S, R346K, K417T/N, K444N, G446V, L452R/Q, L455F, G476S, S477I, T478K, V483F, E484K/Q, F490S and S494L/P. These mutations are predicted to adversely affect the activity of sotrovimab, letesevimab, imdevimab, bamlanivimab, casirivimab and CT-P59. However, we observe that none of these mutations will lead to the loss of stabilizing interactions or will give rise to steric clashes at the P4A2 Fab:RBD interface. As a result, these mutations in the RBD will probably not reduce the ability of P4A2 to neutralize the SARS-CoV-2 (S9 Fig) [5]. Recently new lineages of Omicron have emerged namely BA.2, BA.2.75, BA.2.12.1, BA.3, BA.4 and BA.5 (S10 Fig) [6]. The 486Phe residue in BA.4 and BA.5 is substituted with Val. Since the side chain of Val is also hydrophobic P4A2 may retain substantial affinity for Spike-RBD. Also as expected, F486V results in reduction of affinity of spike-RBD for ACE2 receptor, but the R493Q reversion mutation reinstates the affinity [7]. R493Q may also attenuate the possible reduction in affinity of spike-RBD for P4A2 due to F486V since the shorter Gln side chain will be closer to the paratope surface and can form new hydrogen bonds with residues of CDR L1 of P4A2. P4A2 may therefore, retain significant capacity to bind to spike protein from BA.4/5 and further studies are required to ascertain the ability of P4A2 to neutralize these strains.

The comparison of P4A2 with other known broadly neutralizing antibodies shows that epitopes of 87G7, 510A5, Cov2-2196, NCV2SG48, NCV2SG53, S2E12, S2K146 and ZWD12 show varying degrees of overlap with that of P4A2. Among these mAbs, P4A2 is the only one that forms multiple interactions with its cognate epitope on spike-RBD which is composed of residues that are critical for interaction with ACE2 (S11 Fig and S2 Table) [8–21]. Overall, our data suggests that P4A2 may represent a viable therapeutic option which is required to reduce the impact of the COVID19 pandemic on human health across the globe. Overall, our data shows that P4A2 alone is efficacious in providing protection from the tested VOCs and this ability may probably be retained for majority of the future VOCs. Humanized P4A2 mAb may

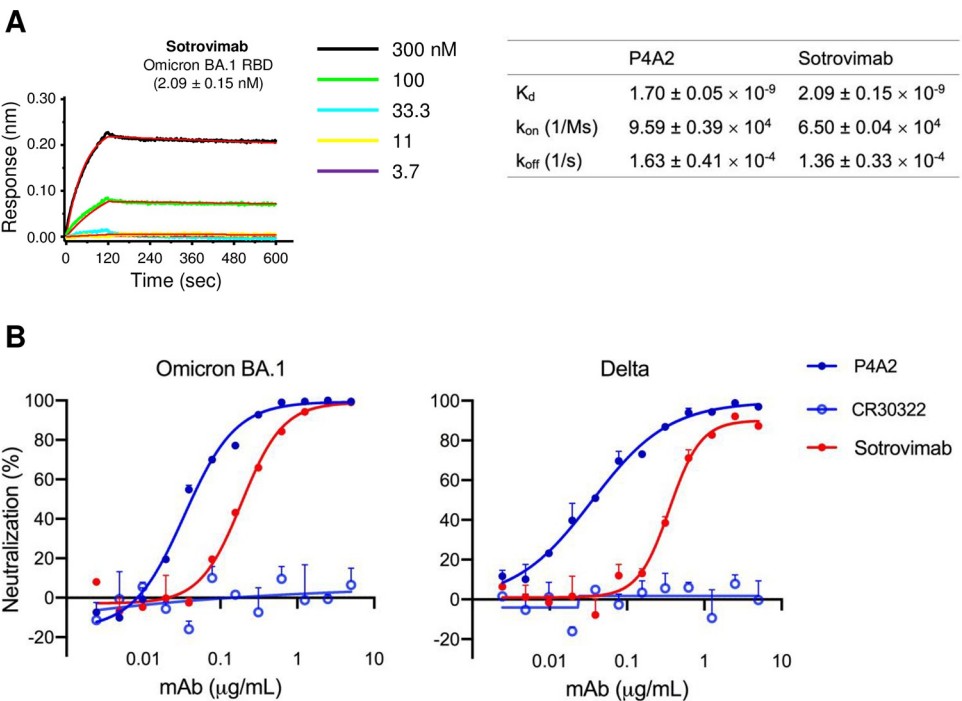

**Fig 3. Neutralization potential of P4A2 mAb against Omicron BA.1 variant.** (A) Sotrovimab was immobilized on anti-human Fc biosensor and tested using three-fold serial dilutions of RBD (starting with 300 nM and going down to 3.3 nM; the five concentrations tested are indicated). Data shown is after the reference was subtracted and aligned using Octet Data Analysis software v11.1 (Forte Bio). Curve fitting was done with a 1:1 binding model, and $k_{on}$, $k_{off}$ and $K_d$ values were calculated using a global fit. (B) Neutralization of authentic SARS-CoV-2 BA.1 and Delta by P4A2 and Sotrovimab was determined using focus-reduction neutralization assay. CR3022 mAb was used as the negative control. Neutralization assays were done in duplicates and repeated at least three times.

be used alone or in conjunction with other non-competing antibodies as an effective prophylactic or therapeutic strategy against current and future variants of SARS-CoV-2. Our studies with the P4A2 mAb also reinforce the idea that therapeutic molecules that bind to regions on the target protein that are critical for natural function will generally be less vulnerable to loss of sensitivity due to mutations in the target protein.

# Materials and methods

## Ethics statement

All the mice were procured from the Small Animal Facility, THSTI, Faridabad. All animal experiments were conducted in accordance with the guidelines for the care and use of laboratory animals as promulgated by Committee for the Purpose of Control and Supervision of Experiments on Animals (CPCSEA), Government of India and adopted by the Institutional Animal Ethics Committee of THSTI (IAEC Project/Protocol No: THSTI-IAEC-146, IAEC-160). The approval of the Institutional Biosafety Committee (approval # THS-354/2021) and Department of Biotechnology Review Committee on Genetic Manipulation (RCGM approval #: BT/IBKP/137/2022) were taken before commencing work.

## Viruses and antibodies

SARS-CoV-2 to B.6 and delta lineage viruses were isolated as described previously [22,23]. Leo Poon provided SARS-CoV-2 Omicron isolate (sub-lineage BA.1) [24]. Vero E6 or Calu-3 cells

were used to propagate SARS-CoV-2 variants. Dr. Raiees Andrabi (Scripps Research Institute, USA) generously provided the full-length spike proteins of SARS-CoV-1, SARS-CoV-2, MERS, HKU1, OC43, NL63 and 229E. CR3022 antibody was purchased commercially (Sino Biologicals). II62 (isotype IgG), a non-neutralizing antibody was used from our previous study [25]. The pseudotyped viral stocks of all seven alpha and beta coronaviruses were prepared as described in our previous study [26]. We used B.1.1.7 (Alpha), B1.1.351 (Beta), B.1.617.1 (Kappa), B.1.617.2 (Delta) and B.1.1.529 (Omicron, BA.1) terminology throughout our manuscript.

## Expression and purification of antibody and RBD protein

A synthetic codon optimized (for mammalian cells) nucleotide sequence of RBD-His (wildtype), spike protein (wildtype), a point mutant of RBD-His (N501Y) protein (variant B.1.1.7) and RBD corresponding to other variants from SARS-CoV-2 were cloned in pcDNA3.1 plasmid vector. The constructs were expressed and produced in Expi 293 F mammalian expression system [11]. Briefly, the cells were transiently transfected with the plasmids. The supernatant was collected after 5–6 days and the soluble protein was purified using Ni-NTA affinity chromatography (Qiagen, Germany).

## Generation of hybridoma for anti-RBD (N501Y) murine monoclonal antibodies

Six to eight week old, female BALB/c mice were immunized intramuscularly with purified RBD (N501Y) protein (30 μg in 100 μL PBS per animal) along with Quil A adjuvant (InvivoGen, USA). Mice were boosted thrice with the purified protein (30, 15 and 7.5 μg in 100 μl PBS per animal), along with Quil A adjuvant. Sera samples from mice were collected three days after the first and second booster. The mouse with the highest titer of serum cross-neutralizing antibodies, was given a final booster injection 4 days before the spleen was aseptically removed. Splenocytes were utilized in the generation of hybridomas using ClonaCell-HY Hybridoma Generation Kit (STEMCELL Technologies, USA), following the manufacturer's protocol. The well-adapted antibody secreting clones were propagated in tissue culture flasks and vials were stored in liquid nitrogen for future use. Hybridoma clones secreting anti-RBD antibodies were further screened by ELISA.

## ELISA

For the screening of hybridomas and heat inactivated mice sera (dilution starting from 1:100 to 1:218, 700)/ purified mAbs (5 to 0.002 μg mL$^{-1}$), ELISA plates were coated with recombinant RBD (N501Y) protein (1 μg mL$^{-1}$; 100 μL per well). Plates were blocked with 5% non-fat milk and hybridoma culture supernatant or purified mAbs were added in three-fold serial dilutions. Following incubation for 1 h at room temperature, HRP conjugated goat anti-mouse secondary antibody (Jackson Immunoresearch, USA; diluted 1 in 2500) was added to each well. In the case of titration experiments, purified mAbs (100 μL per well) were added as three-fold serial dilutions starting with 5 μg mL$^{-1}$. Standard protocols of blocking (5% skimmed milk in PBS) and washing of ELISA plates (4 times with PBS-0.05% Tween-20) were followed as described previously [27,28]. All the experiments were repeated at least three times in triplicates.

## Purification of anti-RBD (N501Y) murine monoclonal antibodies

Serum-free medium designated for mAb production was used to propagate the hybridoma cells ($0.5 \times 10^5$ cells mL$^{-1}$) in a T175 tissue culture flask in order to purify the antibodies. For

scale up of antibody production, a WHEATON CELLine flask was used for hybridoma culture, following the manufacturer's instructions. Protein G agarose resin (G-Biosciences) was used to purify the anti-RBD IgG mAbs from the hybridoma culture supernatant. Five column volumes of PBS were used to wash the beads in the column. Two to three column volumes of 0.1 M glycine (pH 2.5) was added to elute the antibodies, followed by neutralization with 1M Tris-HCl (pH 8.0). The purified antibodies were dialyzed against PBS three times using dialysis tubing (Thermo Fisher Scientific; MWCO = 10 kDa), and concentrated with a 50 kDa cut-off Amicon Ultra-15 centrifuge unit (Millipore). A 0.2 μm syringe filter (MDI, India) was used to filter the antibody solutions before they were used in experiments. NanoDrop spectrophotometer was used to estimate the protein concentration and purified IgG mAbs were analyzed using 12% Tris-Glycine-SDS-PAGE.

## Cytopathic effect-based neutralization assay

Initial screening of heat-inactivated mice serum samples and hybridoma culture supernatants was performed as described [25,29] with slight modification. Briefly, heat-inactivated serum or purified mAbs were serially diluted two or four times and mixed with 100 $TCID_{50}$ of SARS-CoV-2 isolates. The serum or mAb mixture was transferred to the Vero E6 monolayer seeded in a 96-well plate in triplicate and incubated for 1 h. The cell surface was washed with serum-free medium and replenished with fresh complete medium. The plate was further incubated for 72 h at 37°C in a humidified $CO_2$ incubator. The cells were observed for the absence of viral cytopathic effect and was used as indicative of neutralization. The neutralization titer was defined as the dilution at which no cytopathic effect was seen. All the experiments were repeated at least three times in triplicates.

## Live virus focus reduction neutralization assay

Virus neutralization assay was performed as described previously by our group [30]. Inhibitory concentration was assessed by focus reduction neutralization assay using SARS-CoV-2 Delta (GenBank accession no.: MZ356904.1) and Omicron BA.1 (GISAID accession no.: EPI_ISL_8764350) variant. The virus neutralization assay was performed in Vero E6 cells. Cells were incubated for 24 h for Delta and 32 h for the Omicron variant. The virus stock was propagated in Calu-3 cells (American Type Culture Collection, USA). Control purified IgG was used as negative control. All the experiments were repeated at least three times in triplicates.

Vero E6 cells were maintained in Minimal Essential Medium (MEM) containing 10% heat-inactivated fetal bovine serum (FBS), 1% penicillin-streptomycin solution, 1% Non-Essential Amino Acid seeded in a 96-well plate (25,000 cells per well) and incubated overnight at 37°C with 5% $CO_2$. Reagents were two-fold serially diluted (75 μL + 75 μL) in dilution medium (MEM with 1% FBS) from 5 to 0.00244 μg $mL^{-1}$. Predefined virus dilution (75 μL) was added and incubated for 1 h at 37°C with 5% $CO_2$. For the quality control step, pre-defined SARS-CoV-2 neutralizing antibodies positive serum was used. Virus dilution was determined to get approximately 60–200 foci with cell control. After neutralization incubation virus-reagent mixture was added over cell monolayer for virus adsorption and incubated for 1 h at 37°C with 5% $CO_2$. Overlaying medium was added after adsorption incubation and cells were fixed after 24 h. In the case of BA.1, cells were fixed after 28 h of incubation. Cells were permeabilized using 100 μL of IMF buffer and incubated at room temperature for 20 min. Cells were stained with the addition of 100 μL of anti-nucleocapsid antibody at 1:2000 dilution for 1 h, followed by 100 μL of 1:500 dilution of Alexa flour 488 conjugated donkey anti-mouse IgG secondary antibody. After incubation, cells were washed thrice with PBS and developed foci were

quantified by AID iSpot Reader (AID GmbH, Germany) using AID EliSpot 8.0 iSpot software. Using AID raw file to find $IC_{50}$ by using GraphPad Prism (Version 9.3.1) and graphs were plotted with log(inhibitor) versus normalized response—Variable slope parameter. The purified P4A2 antibody was two-fold serially diluted starting from 5 to 0.00244 μg mL$^{-1}$. The virus neutralization assay was performed in Vero E6 cells. Cells were incubated for 24 h for Delta and 32 h for BA.1 variant. The virus stock was propagated in Calu-3 cells (American Type Culture Collection, USA). Control purified IgG was used as the negative control. All the experiments were repeated at least three times in triplicates.

## Pseudovirus-based neutralization assay

Full length spike proteins of all alpha and beta coronaviruses were co-transfected in $1.25 \times 10^5$ HEK293T cells, with helper plasmid expressing firefly luciferase, an HIV-1 backbone and for SARS-CoV-1 and -2, serine protease TMPRSS2 (CMV-Luc, RΔ8.2 backbone plasmid, pTMPRSS2). After 68–72 h culture supernatant was collected and stored at -70˚C. To access the cross-neutralization potential, three-fold serial dilution (starting at 10 μg mL$^{-1}$) of P4A2 was performed. The serially diluted P4A2 antibody was mixed with respective pseudotyped viruses for 1 h at 37˚C. Pseudovirus/ P4A2 combinations were added to 293T-ACE2 cells pre-seeded (24 h earlier) at 20,000 cells per well. After 48–72 h, relative luminescence unit (RLU) was measured on a luminometer. The percent reduction in neutralization was measured as ratio of RLU readout in the presence of P4A2 normalized to RLU readout in the absence of P4A2. Four-parameter logistic regression was used to calculate the half maximum inhibitory concentrations ($IC_{50}$) (GraphPad Prism version 8.3). II62 and CR3022, mAbs against SARS-CoV-2, were used as assay controls. All the experiments were repeated at least three times in triplicates.

## Immunofluorescence microscopy

Vero E6 (25,000 cells per well) seeded in 96-well plate. Virus suspensions at the indicated MOIs were added. After one-hour incubation virus suspensions were removed and overlaying medium (1.5% CMC) was added. Cells were fixed with 7.4% formaldehyde after 24 hours incubation and left overnight. Cells were washed with PBS thrice and permeabilized with 100 μL permeabilization buffer (20 mM HEPES, pH 7.5, 0.1% Triton X-100, 150 mM NaCl, 5 mM EDTA, 0.02% sodium azide) at room temperature for 20 min. Cells were stained with P4A2 as primary antibody (diluted 1 in 2000) at room temperature for 1 h. Secondary Alexa 488 conjugated antibody (anti-mouse Alexa 488, cat no.: A21206, Invitrogen; 1:500) added after washing thrice with buffer. Micro-plaques were estimated using AID iSpot Analyzer (Autoimmun Diagnostika GmbH) and foci were counted using AID ELISPOT 8.0 software.

To check the binding of P4A2 with Delta and BA.1 infected Calu-3 cells (90,000 per well) were seeded in 8-well chamber slides. Cells were incubated for 48 h at 37˚C in $CO_2$ incubator. DMEM high glucose with 10% (v/v) FBS was removed 24 h later and cells were washed with PBS. Cells were infected at a MOI of 0.5 (DMEM high glucose supplemented with 2% FBS). Delta and BA.1 virus (100 μL per well) was added and incubated at 37˚C on a rocker. The virus was removed and washed twice with PBS. Then, 300 μL of 10% complete DMEM was added and incubated further for 24 h at 37˚C.

## Biolayer interferometry binding assay

Binding assays were carried out using an Octet Red instrument (ForteBio Inc.) using BLI as previously described [25,31]. Briefly, mouse Fc sensors (ForteBio Inc.) were used to capture P4A2 at 10 μg mL$^{-1}$ in $1 \times$ kinetics buffer [PBS, pH 7.4, 0.01% (w/v) BSA and 0.002% (v/v)

Tween-20] and incubated at the indicated concentrations of RBD. Associations and dissociations has been reported, depending on the analyte. Data was analysed using the software Forte-Bio Data Analysis. The starting concentration of RBD was 300 nM followed by three-fold serial dilution. All the experiments were repeated at least three times.

## Purification, crystallization, data collection, refinement and analysis

wA total of 30 mg of P4A2 was digested for Fab preparation. The Fab preparation was performed by Pierce Fab Preparation Kit (Cat. no.: 44985) as per the manufacturer's protocol. The purified Fab and RBD proteins were mixed at a molar ratio of 1:1.7 and incubated overnight at 4˚C. The sample was subjected to size exclusion chromatography on 16/600 Superdex 200 column (Cytiva) in a buffer containing 20 mM HEPES (pH 7.5) and 150 mM NaCl. The peak corresponding to P4A2 Fab:RBD complex was concentrated to 10 mg mL$^{-1}$ and stored at -80˚C by flash freezing. The purified complex was subjected to crystallization trials using commercially available screens and trays were set up using Mosquito Crystallization Robot (TTP Labtech). The hits obtained in different screens were further expanded to produce single crystals which were tested for diffraction using a METALJET X-ray home source (Bruker Inc.). The condition that provided crystals with best diffraction quality was composed of 0.2 M magnesium formate dihydrate and 10% PEG 5KMME. These crystals were frozen with 20% glycerol as cryo-protectant. X-ray diffraction data could be collected to a maximal resolution of 3.0 Å at the automated ID30A-1 beamline in ESRF, France [32]. The diffraction data was processed using IMOSFLM [33] and AIMLESS [34] programs of the CCP4 suite (S3 Table).

The structure was determined by molecular replacement using PHASER [35] and the search model was the STEC90-C11 Fab:RBD complex [36]. This model was subjected to iterative model building and refinement using COOT [37] and PHENIX [38], respectively and the sequence of the STEC90-C11 Fab was slowly changed to that of P4A2. The final R$_{free}$ and R$_{work}$ are 28.2 and 23.0%, respectively. The refined structure was deposited with Protein Data Bank with the accession code 7WVL

The structure was visualized and analysed using PYMOL (Schrödinger Corp.) and the interactions were identified using the CONTACT program of CCP4 [39]. Mutations were created *in silico* in the RBD structure using PYMOL to obtain models of P4A2 Fab bound to RBD corresponding to different SARS-CoV-2 strains and Omicron lineages. These models were subjected to energy minimization using the DESMOND module of the Schrodinger suite (Schrodinger Inc.) and analyzed. Two models of P4A2 bound to spike trimer with the RBD in the up and down conformation were prepared using 7TM0 [40] and 7TOU [41] and these models were also subjected to energy minimization using the DESMOND module. All the figures were prepared using PYMOL.

## Animal protection studies

Eight to ten week old K18-hACE2 transgenic mice were pebbled and randomly allotted to different groups (n = 5) *viz*., infection control and those receiving P4A2 in different cages. The animal experiments and procedures were performed in accordance with the Institutional Animal Ethics Committee (IAEC), Institutional Biosafety Committee (IBSC) and Review Committee on Genetic Manipulation (RCGM) guidelines. In prophylactic treatment, antibody recipient groups were given intraperitoneal infusion of P4A2 one day prior to challenge (day '-1'), except for the control group where PBS was given (no virus challenge). In therapeutic treatment group, the mAb was administered 12 h post-infection.

### Clinical spectrum of SARS-CoV-2 infection

The mouse experiments were done in an Animal Biosafety Laboratory (ABSL)-3. Change in daily body weight, activity and clinical symptoms of all the animals were monitored post-infection. On day 6, all the infected animals were euthanized, the lungs were collected and imaged for gross morphological studies. The right lower lobe of the lung was immersed in a 10% (v/v) neutral formalin solution and subjected to immunohistochemistry analysis. The viral load parameters were analysed using homogenized lung tissues in 2 mL Trizol solution. The homogenates were stored immediately at -80˚C till further use. Blood was drawn from the animals via the retro-orbital vein on days '-1' and '0', and via direct heart puncture after euthanizing the animal. Serum samples were stored at -80˚C for future experiments.

### Quantification of viral load in lung

RNA was isolated from homogenised lung tissues using the Trizol-chloroform technique according to the manufacturer's procedure, and quantified using Nanodrop. The iScript cDNA synthesis kit (Bio-Rad, USA) was used for cDNA synthesis. Briefly, 1 μg total RNA was reverse-transcribed into cDNA. The qPCR was performed on diluted cDNAs (1:5) using the KAPA SYBR FAST qPCR Master Mix (5×) Universal Kit (KK4600) and 7500 Dx real-time PCR equipment (Applied Biosystems, USA). The results were analysed with SDS2.1 software as previously described [42]. For virus load estimation, the CDC-approved SARS-CoV-2 N gene primers 5′-GACCCCAAAATCAGCGAAAT-3′ (forward) and 5′-TCTGGTTACTGC-CAGTTGAATCTG-3′ (reverse) were used as previously described [43]. The copy number of N gene was calculated by using pre-titrated SARS-CoV-2 genomic RNA and expressed as N copy number/ lung mass (mg). To produce the standard curve for absolute quantification, a known copy number of viral RNA was employed as a standard.

### Reverse transcriptase-polymerase chain reaction and nucleotide sequencing

The variable region of the immunoglobulin heavy and light chain transcripts expressed in B cell hybridoma P4A2 was RT-PCR amplified and sequenced following the protocol and primers described previously [44]. Briefly, cDNA was synthesized from 10 to 50 snap frozen hybridoma cells, using a commercially available kit (Qiagen, Germany) with isotype specific antisense primers, each at a concentration of 0.75 μM. The 20 μL reaction was performed at 42˚C for 30 min. Reverse transcriptase was inactivated by incubating at 95˚C for 3 min. The nested PCR amplification was performed using Q5 DNA polymerase (New England Biolabs, USA). The cDNA (4 μL) was used as template in a 50 μL first round PCR which comprised of external antisense primer (0.25 μM) and a cocktail of $V_H$ (or $V_L$ as the case may be) family specific external sense primers, each at a final concentration of 0.1 μM, 1 × Q5 DNA polymerase buffer, dNTPs (200 μM) and Q5 DNA polymerase (0.5 U) as recommended by the manufacturer. Two microliter of the first round PCR product was used as template in a 50 μL second round nested PCR following the protocol described above for the first round. The second round PCR product was column purified following the manufacturer's instructions (Invitrogen, USA) and sequenced.

### Sequence analysis

The nucleotide sequence was analyzed using Sequencher (version 5.4.5; Gene Codes, USA) and MacVector (version 17.5.4; MacVector Inc., USA) software. The V, D and J gene segment

assignment was done using IMGT/V-QUEST (https://www.imgt.org/IMGT_vquest/input) [45,46] and IgBlast (https://www.ncbi.nlm.nih.gov/igblast/) [47] using default parameters.

## Supporting information

**S1 Fig. Binding kinetics of mAb P4A2 to RBDs from various VOCs by ELISA and biolayer interferometry (BLI).** (A) P4A2 was immobilized on anti-mouse Fc biosensor and was tested using three-fold serial dilutions of RBD (starting with 300 nM and going down to 3.3 nM; the five concentrations tested are indicated). Data shown is after the reference was subtracted and aligned using Octet Data Analysis software v11.1 (Forte Bio). Curve fitting was done with a 1:1 binding model, and $k_{on}$, $k_{off}$ and $K_d$ values were calculated using a global fit. (B) Cross reactive binding potential (with half-maximal effective concentration, $EC_{50}$) of P4A2 mAb to RBD proteins of different VOCs was tested by indirect ELISA. (C) The epitope specificity of P4A2 was evaluated for epitope competition using BLI. RBD-Fc was captured using anti-human Fc biosensor and saturated with P4A2 and the indicated mAbs at a concentration of 40 µg/ml and unbound P4A2 was washed, followed by incubation with 20 µg/ml of ACE2, P4A2 and P5B3 (binds to a topologically distinct, non-competing epitope and served as a control). No binding signal was observed for P4A2 and ACE2.
(PDF)

**S2 Fig. Mutations in RBD associated with different VOCs do not affect binding to P4A2 Fab.** Using the crystal structure, computational models of P4A2 Fab in complex with the RBD from (A) Beta, (B) Gamma, (C) Delta, (D) Kappa and (E) BA.1 VOCs were generated. These models show that, for all the VOCs, there are no mutations in the residues that interact with the P4A2 through their side-chain and hence these mutations will not adversely impact P4A2 binding. E484 is mutated to Lys, Gln or Ala in some of the VOCs, but it forms interactions with the P4A2 Fab paratope through the backbone atoms and not through the side chain.
(PDF)

**S3 Fig. Immunofluorescence intensity of P4A2 binding to live virus infected Vero E6 cells.** The binding of P4A2 to the spike proteins expressed on the surface of Vero E6 cell infected with various VOCs was assessed by immunofluorescence microscopy. Vero E6 cells were infected at a MOI of 0.1 and 0.01. The number of foci at MOI 0.1 were too numerous to be counted. The foci count for the MOI 0.01 are indicated. P4A2 was used as the primary antibody (diluted 1 in 2000) followed by anti-mouse Alexa 488 as the secondary antibody. The foci were counted using AID EliSpot 8.0 software. Representative images of the experiment performed in triplicates are shown.
(PDF)

**S4 Fig. Immunofluorescence staining of P4A2 binding to live virus infected Vero E6 cells.** Vero E6 cells were infected with Wuhan (A), Delta (B) and BA.1 variant of SARS-CoV-2 at a MOI of 0.1 (C) and 0.01 (D). Cells were fixed with 7.4% formaldehyde 32 h following infection, stained with mAb P4A2 and observed under an immunofluorescence microscope. DAPI was used to stain nuclei. Images were captured at a magnification of 10×. Scale bar represents 100 µm.
(PDF)

**S5 Fig. Relative change in the body mass and gross morphology of lungs excised from uninfected or infected mice showing pneumonitis and inflammation.** (A) Body mass of mice from each group (both prophylactic and therapeutic) was recorded for 6 days post-infection. The Y-axis represents the body mass of the mouse relative to the body mass of the same

mouse recorded on day 0 (normalised to 100) (dpi, days post-infection). (B) Animal challenge experiment performed with prophylactic or therapeutic intervention of P4A2 antibody. Briefly, animals challenged with Wuhan, Kappa, Delta or Beta SARS-CoV-2 strain ($10^5$ pfu/ mice) were given prophylactic (1 day prior to challenge) or therapeutic dose (12 h post challenge) and representative images of the excised lung showing inflammation and pneumonitis. (PDF)

**S6 Fig. Pulmonary pathology of SARS-CoV-2 infected hACE2 mice in prophylactic treatment.** Animals challenged with SARS-CoV-2 with or without P4A2 antibody were euthanized on day 6 post-infection. The left lower lobe of their lung were fixed in 10% formalin solution and used for H & E staining. (A) Representative images of transverse section of the lung showing pneumonitis (magnification = 40×; red arrow), alveolar epithelial injury (blue arrow) and inflammation (black arrow). The stained sections were assessed by blinded-trained histologist on the scale of 0–5 (where 0 represents no feature, while 5 represents the maximum score). (B) The histological scores for each pulmonary pathology was plotted as mean ± SEM. The disease index score was calculated by taking the average score of all pulmonary pathologies (i. e. pneumonitis, alveolar epithelial injury and inflammation). Ther, therapeutic; Pro, prophylactic. NS, not statistically significant. (PDF)

**S7 Fig. Pulmonary pathology of SARS-CoV-2 infected K18-hACE2 mice in therapeutic group.** Lung samples from euthanized animals 6 days post challenge were fixed in 10% formalin solution and stained with H & E. (A) Representative transverse section of the H & E stained lung images at 40× magnification showing pneumonitis (red arrow), alveolar epithelial injury (blue arrow) and inflammation (black arrow). The stained sections were assessed by blinded-trained histologist on the scale of 0–5 (where 0 represents no feature, while 5 represents the maximum score). (B) The histological scores for each pulmonary pathology was plotted as mean ± SEM. The disease index score was calculated by taking the average score of all pulmonary pathologies. Ther, therapeutic; NS, not statistically significant. (PDF)

**S8 Fig. Representative layout and images of focus reduction neutralization assay with P4A2 mAb.** (A) SARS-CoV-2 neutralizing antibodies were used as the positive control [two-fold dilution series starting at 1:20 and ending at 1:640 (highlighted in green)]. Pre-defined virus dilution (virus only control; wells 7 through 11 in rows A and B to get at least 60–200 FFU/well) and only medium (no virus control; added in well 12 in rows A and B) served as the control. P4A2 (rows C and D), Sotrovimab (rows E and F) and CR30322 (rows G and H) as two-fold serial dilutions were used in the experiment. The antibody neutralization assays against SARS-CoV-2 Delta variant (B.1.617.2) (B) and Omicron variant (B.1.1.529) (C) was done similarly. (PDF)

**S9 Fig. Effect of predicted mutations on binding of P4A2 mAb.** (A) Amino acid sequence of the variable regions of heavy and light chain of mAb P4A2 are denoted as P4A2-H and P4A2-L, respectively. Framework regions are shown in black, whereas the complementary determining region 1, 2 and 3 are highlighted in red, blue and green, respectively. (B) The mutations predicted by Maher et al. (2022) [5] are displayed in stick representation and coloured according to element. The spike-RBD, heavy and light chains of P4A2 Fab are shown in magenta, orange and cyan, respectively. None of the predicted mutations of the spike-RBD overlap with the residues that interact with P4A2 Fab and therefore it is possible that these mutations may not reduce the ability of P4A2 to neutralize the corresponding new variants of

SARS-CoV-2.
(PDF)

**S10 Fig. Computational modelling of P4A2 mAb with different Omicron variants.** Computational models of P4A2 Fab in complex with spike-RBD from different Omicron lineages BA.1, BA.2, BA.3, BA.4/5, BA.2.75 and BA.2.12.1.
(PDF)

**S11 Fig. Epitopes of different broadly neutralizing antibodies.** The surface of the spike-RBD is displayed and the epitope for different mAbs are shown in red colour. The mAbs JMB2002 and S3H3 are not shown here because the epitopes for these two antibodies are outside the RBD. The epitopes of 87G7, 510A5, Cov2-2196, NCV2SG48, NCV2SG53, S2E12, S2K146 and ZWD12 showed some overlap with that of P4A2 but only P4A2 mAb possesses a hydrophobic cleft into which the 486Phe residue is buried. Based on available information, P4A2 forms multiple interactions with its cognate epitope on spike-RBD and multiple residues present in this epitope are critical for interaction with ACE2.
(PDF)

**S1 Table. Comparative neutralization potential of P4A2 mAb with the clinical approved or under development therapies.**
(DOCX)

**S2 Table. Comparative epitope analysis of P4A2 mAb with other known ACE2 inhibiting antibodies.**
(DOCX)

**S3 Table. Crystallographic data and refinement statistics for P4A2 Fab:spike-RBD complex.**
(PDF)

## Acknowledgments

We thank Prof. Sudhanshu Vrati, Director, RCB for establishing a collaborative effort on structural aspects of broadly neutralizing mAbs. The following reagent was deposited by the Centre for Disease Control and Prevention and obtained through BEI Resources, NIAID, NIH: SARS-Related Coronavirus 2, Isolate USA-WA1/2020, NR-52281, Isolate USA/CA_CDC_5574/2020, NR-54011 and Isolate hCoV-19/South Africa/KRISP-K005325/2020, (NR-54009, contributed by Alex Sigal and Tulio de Oliveira), Isolate hCoV-19/USA/PHC658/2021 (Lineage B.1.617.2; Delta Variant). The authors would like to thank Dr. Didier Nurizzo for help with data collection at the ID30A-1 beamline of the European Synchrotron Radiation Facility (ESRF).

## Author Contributions

**Conceptualization:** Sachin Kushwaha, Subrata Sinha, Devinder Sehgal, Chandresh Sharma, Amit Awasthi, Pramod Kumar Garg, Deepak T. Nair, Rajesh Kumar.

**Data curation:** Naveen Narayanan, Sonal Garg, Zaigham Abbas Rizvi, Tripti Shrivastava, Sachin Kushwaha, Janmejay Singh, Praveenkumar Murugavelu, Farha Mehdi, Nisha Raj, Shivam Singh, Chandresh Sharma, Deepak T. Nair, Rajesh Kumar.

**Formal analysis:** Naveen Narayanan, Tripti Shrivastava, Sachin Kushwaha, Praveenkumar Murugavelu, Anbalagan Anantharaj, Farha Mehdi, Nisha Raj, Shivam Singh, Guruprasad R. Medigeshi, Pramod Kumar Garg, Deepak T. Nair, Rajesh Kumar.

**Funding acquisition:** Sonal Garg, Amit Awasthi, Pramod Kumar Garg, Deepak T. Nair, Rajesh Kumar.

**Investigation:** Hilal Ahmad Parray, Naveen Narayanan, Zaigham Abbas Rizvi, Tripti Shrivastava, Sachin Kushwaha, Janmejay Singh, Anbalagan Anantharaj, Farha Mehdi, Nisha Raj, Shivam Singh, Jyotsna Dandotiya, Adarsh K. Chiranjivi, Nitesh Mishra, Kamini Jakhar, Sudipta Sonar, Shailendra Mani, Sankar Bhattacharyya, Supratik Das, Kalpana Luthra, Gaurav Batra, Devinder Sehgal, Guruprasad R. Medigeshi, Chandresh Sharma, Amit Awasthi, Pramod Kumar Garg, Deepak T. Nair, Rajesh Kumar.

**Methodology:** Hilal Ahmad Parray, Naveen Narayanan, Sonal Garg, Zaigham Abbas Rizvi, Tripti Shrivastava, Janmejay Singh, Anbalagan Anantharaj, Asha Lukose, Deepti Jamwal, Samridhi Dhyani, Nitesh Mishra, Kamini Jakhar, Sudipta Sonar, Anil Kumar Panchal, Shailendra Mani, Sankar Bhattacharyya, Supratik Das, Kalpana Luthra, Gaurav Batra, Devinder Sehgal, Guruprasad R. Medigeshi, Chandresh Sharma, Amit Awasthi, Pramod Kumar Garg, Deepak T. Nair, Rajesh Kumar.

**Project administration:** Rajesh Kumar.

**Resources:** Sandeep Kumar, Sanjeev Kumar, Manas Ranjan Tripathy, Shirlie Roy Chowdhury, Shubbir Ahmed, Sweety Samal, Kalpana Luthra, Gaurav Batra, Devinder Sehgal, Guruprasad R. Medigeshi, Chandresh Sharma, Amit Awasthi, Pramod Kumar Garg, Deepak T. Nair, Rajesh Kumar.

**Supervision:** Pramod Kumar Garg, Deepak T. Nair, Rajesh Kumar.

**Validation:** Guruprasad R. Medigeshi, Amit Awasthi, Pramod Kumar Garg, Deepak T. Nair.

**Writing – original draft:** Deepak T. Nair, Rajesh Kumar.

**Writing – review & editing:** Subrata Sinha, Devinder Sehgal, Guruprasad R. Medigeshi, Amit Awasthi, Pramod Kumar Garg, Deepak T. Nair, Rajesh Kumar.

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
