## [Decision Letter · Decision Letter 0]

12 Sep 2022

Dear Dr Kumar,

Thank you very much for submitting your manuscript "A broadly neutralising monoclonal antibody overcomes the mutational landscape of emerging SARS-CoV2 variant of concerns" for consideration at PLOS Pathogens. As with all papers reviewed by the journal, your manuscript was reviewed by members of the editorial board and by several independent reviewers. The reviewers appreciated the attention to an important topic. Based on the reviews, we are likely to accept this manuscript for publication, providing that you modify the manuscript according to the review recommendations.

All 3 reviewers stated that the authors satisfactorily addressed previous comments, and that this work is highly significant towards tackling SAR-CoV-2 variants. Reviewers 2 and 3 provided helpful comments to increase manuscript quality (both in terms of the text and figures). Thank you in advance for addressing each reviewer's comments.

Sincerely,

Sujan Shresta

Associate Editor

PLOS Pathogens

Kanta Subbarao

Section Editor

PLOS Pathogens

Kasturi Haldar

Editor-in-Chief

PLOS Pathogens

orcid.org/0000-0001-5065-158X

Michael Malim

Editor-in-Chief

PLOS Pathogens

orcid.org/0000-0002-7699-2064

All 3 reviewers stated that the authors satisfactorily addressed previous comments, and that this work is highly significant towards tackling SAR-CoV-2 variants. Reviewers 2 and 3 provided helpful comments to increase manuscript quality (both in terms of the text and figures). Thank you in advance for addressing each reviewer's comments.

Reviewer Comments (if any, and for reference):

Reviewer's Responses to Questions

**Part I - Summary**

Reviewer #1: The candidate has taken lot of effort to address the concerns of all the reviewers and to me it is highly satisfactory. May be they can comment (in conclusion) more on new variants of covid and their potential to get neutralized by their MCA but this is extremely minor point

Reviewer #2: With one minor exception, the authors have responded fully and constructively to the comments I made at the initial review. The main thing I would like to see improved in the current version is the description of the methods for the focus reduction neutralization assay. This request is prompted by the new image of the omicron-infected cells in Figure S3. I had previously asked if a better image were available, and the authors have kindly provided one. The new image confirms the impression that the omicron variant produces fewer foci than the other variants. It would therefore make sense to describe the methods for the focus reduction neutralization assay in sufficient detail that a reader could judge whether the very different kinetics of the infection were handled in a way that meant that the IC50 for this variant and the others are comparable. Ideally, images of the neutralization plates could be provided containing at least one series for each of the agents shown on the plot, along with a description of how the parameters of infection were measured and analyzed.

Also on the theme of comparability, I believe that the comparison shown in S8 between experimentally determined IC50 for P4A2, with values from the literature for other agents is of rather limited value presented as it stands, because of the very different values obtained by different labs with nominally similar assays. The experimental comparison between P4A2 and sotrovimab provided by the authors in S9 is much more informative. If these figures S8 and S9 could be combined, so that inter-laboratory variation in the IC50 measured for the same agent were immediately obvious, it would aid intepretation of both figures. S9 is such an important figure that I don't know why it was not included in the main figures, for example in Figure 1.

Regarding my previous comment about the label on the vertical axis of figures 2C and 2E, despite the authors, response, I stand by my initial opinion, and extend it to S5A. To me, a change in body mass of 100% means that the body mass increased or decreased by 100%, suggesting that on day 1 the animals doubled their weight, by day 2 had tripled it, and so on. I am not going to bicker about this - any reasonable person familiar with the lives of mice will understand the figure even if the current format is retained.

The language in the original manuscript was generally, in my opinion, clear and of good quality, while the passages that have been added in response to reviewers' comments are rather more variable in quality and would benefit from systematic reading for grammar.

Reviewer #3: The authors report the identification of a broadly neutralizing antibody derived via immunization of mice with the SARS-CoV-2 Alpha RBD. Structural analysis demonstrates this antibody binds to the upper, outer face of the RBD, in a manner that is not impacted by mutations found in VOCs. Therapeutic and protective efficacy for this antibody, P4A2, are described for a variety of SARS-2 CoV variants. Overall, the paper provides sound biochemical, structural and in vivo analyses of this antibody. The authors have, for the most part, addressed all concerns presented during the initial review. There are several additional points that should be addressed to increase the clarity of both text and figures.

**Part II – Major Issues: Key Experiments Required for Acceptance**

Reviewer #1: No major issues - accepted in the present form

Reviewer #2: I don't envisage that additional experiments would change the conclusions.

Reviewer #3: No major issues are noted.

**Part III – Minor Issues: Editorial and Data Presentation Modifications**

Reviewer #1: None

Reviewer #2: already addressed above.

Reviewer #3: 1. The manuscript would generally benefit from editing to improve language use, construction and clarity. There are a number of typos, misplaced or mis-included words throughout the main text and figure legends. For example:

a. Line 144: “nearly highly reduced” is awkward and I am not clear what the authors intend to say – “nearly X-fold reduced” or perhaps “highly reduced” ?

b. Line 152: should predicated be predicted? If the authors intended to use predicated, I find it difficult to understand.

c. Line 156-157: “This might be one probable reason that both antibodies though having the similar affinity to RBD but shows slightly different functional readouts in terms of neutralization (Fig. S8 and S9)” is very difficult to follow. The authors state that Sotrovimab can neutralize the Omicron variant, but with “significantly lesser potency than P4A2.” But then later say “show slightly different functional readouts”. What do the authors mean by “functional readouts” and how does this relate the significantly lesser potency yet also only slightly different function?

2. Line 161: “the unique mode of RBD recognition employed by P4A2 ensures that this mAb is be able to neutralize new variants that may arise in the future” is too strong of a statement. The virus will continue to evolve and there is no assurance that residues contained within the PA42 epitope will not be mutated in future variants. Indeed, Omicron variant BA.4 is now dominant in many parts of the world and has an F486V mutation, which is a central amino acid in the P4A2 epitope. The authors should discuss (or better yet test) the impact this has on the neutralization profile of P4A2.

3. The authors refer to the use of the “Omicron” VOC throughout the text and note in the methods that it is the B.1.1.529 lineage. However, given the rapid expansion of Omicron-based lineages, it would be useful to clarify the mutations in the virus used and indicate the most closely related sub-lineage that the virus used in the studies corresponds to (i.e. BA.1 or otherwise).

4. I am also not clear as to how the mode of recognition is unique. Several groups have classified RBD-binding antibodies. Does P4A2 fall into one of these defined groups or is the mode truly different? Likewise, many antibodies that bind in a similar location as P4A2 can bind whether the RBD is up or down. Expansion on the uniqueness or removal of this qualifier would help clarify.

5. Figure 1A and B: the authors should include the standard deviation values in these tables that reflect the thrice-repeated measurements.

6. Figure 1I: the Y-axis label is cut off.

7. Figure 1I and J – what do the error bars indicate? Are these representative graphs with error bars for the technical triplicate measurements or a compilation of all three independent experiments?

8. Figure 2E: the color scheme, combined with the line and shape choices make it difficult to understand which data points correspond to which conditions. This will be particularly difficult for readers with red-green color blindness. The authors should consider using different shapes and line styles and make these more apparent in the legend.

9. Figure 2F: the legend states that the data is presented as the Log10 N copy number, but the axis is labeled as “Viral load (AU)” and shows powers of 10. The authors may mean to say the axis is shown on a log scale. If not, the values should be simply 1, 6, 8, etc not 101, 106, 108 and the axis label should be “Log Viral load (AU)”. I am also not familiar with what “AU” indicates and how this relates to copy number. There are also some typos in the legend.

10. Figure S1B: Include std deviation values.

11. Figure S1C: Legend states:

“RBD-Fc was captured using anti-human Fc biosensor and saturated with P4A2 indicated mAbs at a concentration of 40 μg/ml and unbound P4A2 was washed, followed by incubation with 20 μg/ml of ACE2, P4A2 and P5B3 (binds to a topologically distinct, non-competing epitope and served as a control).”

The experimental set up looks like only P4A2 was included as the first antibody (not P4A2 indicated mAbs). P5B3 was included as a control (presumably identified in this same discovery campaign?) but what is P1A2?

12. Figure S2: There are many structural models of the RBDs and Spikes from VOCs. Did the authors compare the models they obtained from their simulation studies to those models? Several groups have reported small, but sometimes significant variations in amino acid mainchain and sidechain orientations amongst VOCs, particularly as it relates to antibody binding. Modeling may not accurately reflect the position of the residues modeled here.

13. Figure S5: what do the circles indicate?

14. Figure S6: The authors should indicate the model and parameters used for assessment of statistical significance in this figure, as well as any other figure with significance reported.

15. Figure S10: While it is true that none of the residues modeled appear to perturb the binding interface, the statement “None of the mutations perturb the interaction between the P4A2 Fab and the Spike-RBD and therefore will not adversely affect the ability of the mAb to neutralize the SARS-CoV-2 virus” is too presumptive without empirical assessment through structural or in vitro analysis. The authors should also include residues identified in newly emergent Omicron lineages, such as BA.2 and BA.4, particularly the F486V mutation.

16. Figure S11:

a. The legend states: “The mAbs JMB2002 and S3H3 are now shown here ...” Do the authors mean to say these mAbs are NOT shown here (as they are not).

b. The omission of Sotrovimab is interesting since the authors specifically show data for this antibody. Is there a reason it was not included?

c. The figure would benefit from a reorganization that groups the RBDs by their orientation.

17. ELISA section of Methods: ug/mL-1 should be changed to ug/mL or ug mL-1.

PLOS authors have the option to publish the peer review history of their article (what does this mean?). If published, this will include your full peer review and any attached files.

Reviewer #1: **Yes: **Akhil Chandra Banerjea

Reviewer #2: No

Reviewer #3: No

Figure Files:

Data Requirements:

Reproducibility:

References:

---

## [Editor Report · Decision Letter 1]

8 Nov 2022

Dear Dr Kumar,

We are pleased to inform you that your manuscript 'A broadly neutralising monoclonal antibody overcomes the mutational landscape of emerging SARS-CoV2 variant of concerns' has been provisionally accepted for publication in PLOS Pathogens.

Best regards,

Sujan Shresta

Associate Editor

PLOS Pathogens

Kanta Subbarao

Section Editor

PLOS Pathogens

Kasturi Haldar

Editor-in-Chief

PLOS Pathogens

orcid.org/0000-0001-5065-158X

Michael Malim

Editor-in-Chief

PLOS Pathogens

orcid.org/0000-0002-7699-2064
---

## [Editor Report · Acceptance letter]

8 Dec 2022

Dear Dr Kumar,

We are delighted to inform you that your manuscript, "A broadly neutralising monoclonal antibody overcomes the mutational landscape of emerging SARS-CoV2 variant of concerns," has been formally accepted for publication in PLOS Pathogens.

Best regards,

Kasturi Haldar

Editor-in-Chief

PLOS Pathogens

orcid.org/0000-0001-5065-158X

Michael Malim

Editor-in-Chief

PLOS Pathogens

orcid.org/0000-0002-7699-2064